# *Arabidopsis* sepals: A model system for the emergent process of morphogenesis

Adrienne H. K. Roeder

Section of Plant Biology, School of Integrative Plant Science and Weill Institute for Cell and Molecular Biology, Cornell University, Ithaca, New York, USA

## Review Article

**Author for correspondence:**
Adrienne H. K. Roeder,
E-mail: ahr75@cornell.edu

### Abstract

During development, *Arabidopsis thaliana* sepal primordium cells grow, divide and interact with their neighbours, giving rise to a sepal with the correct size, shape and form. *Arabidopsis* sepals have proven to be a good system for elucidating the emergent processes driving morphogenesis due to their simplicity, their accessibility for imaging and manipulation, and their reproducible development. Sepals undergo a basipetal gradient of growth, with cessation of cell division, slow growth and maturation starting at the tip of the sepal and progressing to the base. In this review, I discuss five recent examples of processes during sepal morphogenesis that yield emergent properties: robust size, tapered tip shape, laminar shape, scattered giant cells and complex gene expression patterns. In each case, experiments examining the dynamics of sepal development led to the hypotheses of local rules. In each example, a computational model was used to demonstrate that these local rules are sufficient to give rise to the emergent properties of morphogenesis.

## 1. Morphogenesis as an emergent process

How does a flower develop from a few cells to its final beautiful and elaborate shape? Morphogenesis is the development of size, shape, structure and form, which has long fascinated developmental biologists. Morphogenesis is complex because it is an inherently emergent process. Complex systems science defines emergent processes as those in which individual parts (e.g., cells) interact following local rules in such a way that they give rise to new properties in the whole entity (e.g., flower), which are not found in the individual parts (Krakauer, 2019). A classic example of an emergent process is the interaction of water molecules, which do not follow a blueprint, yet freeze into beautiful snowflakes. In morphogenesis, the cells grow, divide and interact with their neighbours and through these local interactions they give rise to a flower with the correct size and shape, which blooms. Emergence takes place across scales: from water molecule to snowflake, or from cells to organs, for example. The relatively simple interactions among small-scale parts give rise to a pattern or form at larger scales that are not an immediately obvious output of the rules.

In the early 1990s, Kaplan and Hagemann strenuously argued that plant morphogenesis must be understood from the perspective of the whole organism (organismal theory) and could not be understood as the aggregate actions of the cells (cell theory) (Kaplan, 1992; Kaplan & Hagemann, 1991). Today, their argument can be reinterpreted as a case for the emergent nature of morphogenesis, namely that the whole organism has properties that are not present and cannot be easily predicted from the cells. Over the last 30 years, our capabilities have increased to the point where we can now begin to elucidate the cellular-scale mechanisms giving rise to emergent properties in the organ and organism, which is the subject of this review.

In plant biology, the generation of spiral phyllotactic patterns of organs around the shoot apical meristem is one of the best examples of an emergent process in which we have some understanding of the local rules (Figure 1a). Decades of experiments have shown that new organ primordia form on the flanks of the shoot apical meristem where the plant hormone auxin concentrates (Heisler et al., 2005; Reinhardt et al., 2000; 2003). Auxin is effluxed (pumped) out of a cell into the cell wall by the PIN1 polar auxin efflux transporter (Okada et al., 1991).

Each cell only 'knows' itself and its local environment, and has no information about where the last primordium occurred across the meristem, and thus no idea where the next primordium should form. Using computational modelling, plant biologists have shown that the spiral pattern of primordia forming around the meristem can emerge from a simple local rule together with growth and cell division (Figure 1a). This simple local rule is that each cell orients its PIN1 protein towards its neighbouring cell that already has the highest auxin concentration (Figure 1a; Jönsson et al., 2006; Smith et al., 2006). Thus, the cell pumps its auxin up the auxin gradient to increase its neighbour's auxin concentration. It is not immediately obvious that this simple rule together with growth would give rise to the continuous generation of a spiral pattern of auxin maxima, yet computational modelling shows it does. Of course, this is only the beginning of our understanding and many questions as well as competing models remain to be addressed (Galván-Ampudia et al., 2020). How does a cell know which neighbour has the highest auxin concentration and polarise its PIN in that direction? Mechanical interactions between neighbouring cells may be central to answering this question (Heisler et al., 2010; Nakayama et al., 2012). Local cell wall growth causes plasma membrane tension which promotes exocytosis of PIN1 to the membrane; whereas membrane fluidity is associated with endocytosis and removal of PIN1 from the membrane (Nakayama et al., 2012). Mechanical signals cause transient increases in calcium concentration, which is required for PIN1 repolarization (Li et al., 2019). Thus, PIN1 may polarise towards the fastest-growing neighbouring cell, which is presumably the one with the highest auxin concentration. Additional players are involved in robustness (i.e., reproducibility in the face of perturbations) of the timing of formation of primordia on the meristem (Besnard et al., 2014).

As is evident in the preceding example, elucidating emergent processes requires quantitative analyses together with computational modelling, a computational morphodyanmics approach (Chickarmane et al., 2010; Roeder et al., 2011). Implementing the small-scale local rules in a computational model is the best way to determine whether simulation of these rules is sufficient to generate the non-intuitive emergent properties in the whole entity in silico. One of the most challenging and creative endeavours for plant biologists is to hypothesise these local rules. These hypotheses are often based on substantial experimentation. Although many kinds of experiments can be informative, for morphogenesis, dynamic imaging experiments in living, growing, developing tissues and organisms are indispensable (Cunha et al., 2012; Hamant et al., 2019). For example, the models of phyllotaxy were developed following extensive analysis of PIN1 polarisation, particularly live imaging of PIN1 dynamics in the developing meristem (Heisler et al., 2005; Reinhardt et al., 2003). Extracting information from these complex imaging datasets requires sophisticated computational image processing (Barbier de Reuille et al., 2015; Fernandez et al., 2010; Roeder, Cunha, Burl, & Meyerowitz, 2012; Wolny et al., 2020). These quantitative data are particularly important for both setting parameters and testing whether simulation of the model matches the biological outcome of morphogenesis or not. Often only the final successful model is included in a publication, but much of the true value in modelling comes from all of the failed models that do not match the data; such models help researchers identify flaws in the design and refine their hypotheses (Harline et al., 2021). In practice, deciphering the emergent processes that drive morphogenesis requires iterative rounds of modelling and dynamic experimentation, with both refining and informing each other (Chickarmane et al., 2010; Roeder et al., 2011).

---

**Fig. 1.** Emergent morphogenesis. For each example of morphogenesis, the local rule is depicted on the left, the model simulation of this rule giving rise to the emergent properties is in the centre, and the final shape and form that are outcomes of the emergent morphogenesis processes are on the right. (a) The spiral pattern of flowers emerging around the *Arabidopsis* inflorescence meristem (right) is produced by the local rule that the PIN1 auxin efflux transporter is polarised towards the neighbouring cell with the highest auxin concentration (left). Simulating the trafficking of auxin based on this rule gives rise to a spiral pattern of auxin maxima which initiate primordia surrounding the growing shoot apex. Simulation image courtesy of Richard S. Smith based on the model published in Smith et al. (2006). (b) The robustness of sepal size and shape in *Arabidopsis* as represented by the overlap in the outlines of wild type sepals (right) emerges from the local rule that each cell varies its growth rate in time and growth also varies between neighbouring cells (rainbow heatmap; in the heatmap each colour represents a different numerical growth ratio with warmer colours representing faster growth in $\mu m^2/\mu m^2$). A simulation of this rule, in which growth rate is determined by stiffness of the mechanical model, produces uniform sepal shapes and sizes (middle). Stiffness is represented as a greyscale heatmap. Images adapted from Hong et al. (2016) with permission from Elsevier. (c) The tapered sepal tip shape in wild type plants emerges from the mechanical feedback loop in which cortical microtubules in the cells reorient to resist mechanical stress (left) combined with the growth gradient which generates mechanical stress at the junction between the fast and slow growing zones. Simulations in which the strength of the mechanical feedback is enhanced (+) make pointier tips, whereas simulations in which the feedback is weakened (−) make rounder tips (middle). The growth gradient is represented as a heat map with fast growth in red and slow growth in blue. These simulations predict the pointier tip of *spiral2* mutants in which mechanical response of microtubules is increased and the rounded sepal tip of *katanin* mutants in which mechanical stress response is reduced (right). Simulation and sepal images reprinted from Hervieux et al. (2016) with permission from Elsevier. (d) The development of the sepal as a lamina or flattened structure (image on the right shows the adaxial side of the sepal) emerges from the mechanical response of cortical microtubules underlying the inner cell walls of the leaf, not the outer epidermal cell wall. Simulation of the growth of a multicellular mechanical model of the sepal or leaf primordium (middle). Simulation 1 represents the initial shape of the model organ surface with cross section below. Simulation 2 of organ growth with no mechanical feedback in either inner our outer cell walls. Note the organ becomes more spherical. Simulation 3 with mechanical feedback on both inner and outer walls. Note that the organ maintains flatness but becomes highly elongated and the predicted microtubule patterns do not match (not shown). Simulation 4 with mechanical feedback only on the outer cell walls. Note that flatness is lost. Simulation 5 with mechanical feedback only on inner and not outer cell walls. Note the model organ becomes a thin laminar structure, best matching the biological organ. Simulation and sepal image reprinted from Zhao et al. (2020) with permission from Elsevier. (e) The pattern of giant cells in the sepal epidermis (false coloured red on the left, scale bars: 100 μm) emerge from the apparently stochastic fluctuations of the ATML1 transcription factor combined with the cell cycle and organ growth. In an individual cell, ATML1 concentration fluctuates (left). This individual giant cell is highlighted with a red line and is superimposed on the traces of ATML1 concentration from the other cells in the sepal shown in grey. The cell cycle stage is indicated with coloured dots: G1 in yellow, G2 in blue, and endoreduplication in red. If ATML1 concentration is above a threshold during G2 phase of the cell cycle, the cell is likely to endoreduplicate (left). Endoreduplicating cells terminally differentiate and do not resume divisions. Simulating these rules in an expanding tissue model generates a pattern of giant cells interspersed between small cells (middle). The giant cells become enlarged and polyploid, growing with the tissue. Simultaneously, the small cells continue to divide, subdividing the same growing tissue into more cells. Images reproduced from Meyer et al. (2017) under the CCBY 4.0 licence. (f) The patterns of gene expression of key regulators on the developing FM (right) emerges from the interactions of these genes in the gene regulatory network (left) and their initial expression patterns. In the gene regulatory network, green nodes are involved in polarity, yellow nodes in identity, blue nodes in outgrowth, and red nodes in meristem. Blue arrowheads indicate positive interactions and blue arrows promote positive interactions, while red arrowheads denote negative interactions and red arrows promote negative interactions. In fact, the gene regulatory network based on interactions from the literature is not sufficient to regenerate many of the gene expression patterns. In AG, AS1, ANT, CUC, AHP6 and REV (expression pattern shown red on a blue background), the addition of one more hypothetical regulatory interaction (listed in red under the model, where ¬ indicates not), comes very close to reproducing the spatial expression pattern (quantified as a BAcc score above each model where 1 represents the perfect match). The combinatory expression patterns of the genes are denoted in the atlas figure on the right by the coloured regions. Images reprinted from Refahi et al. (2021) with permission from Elsevier.

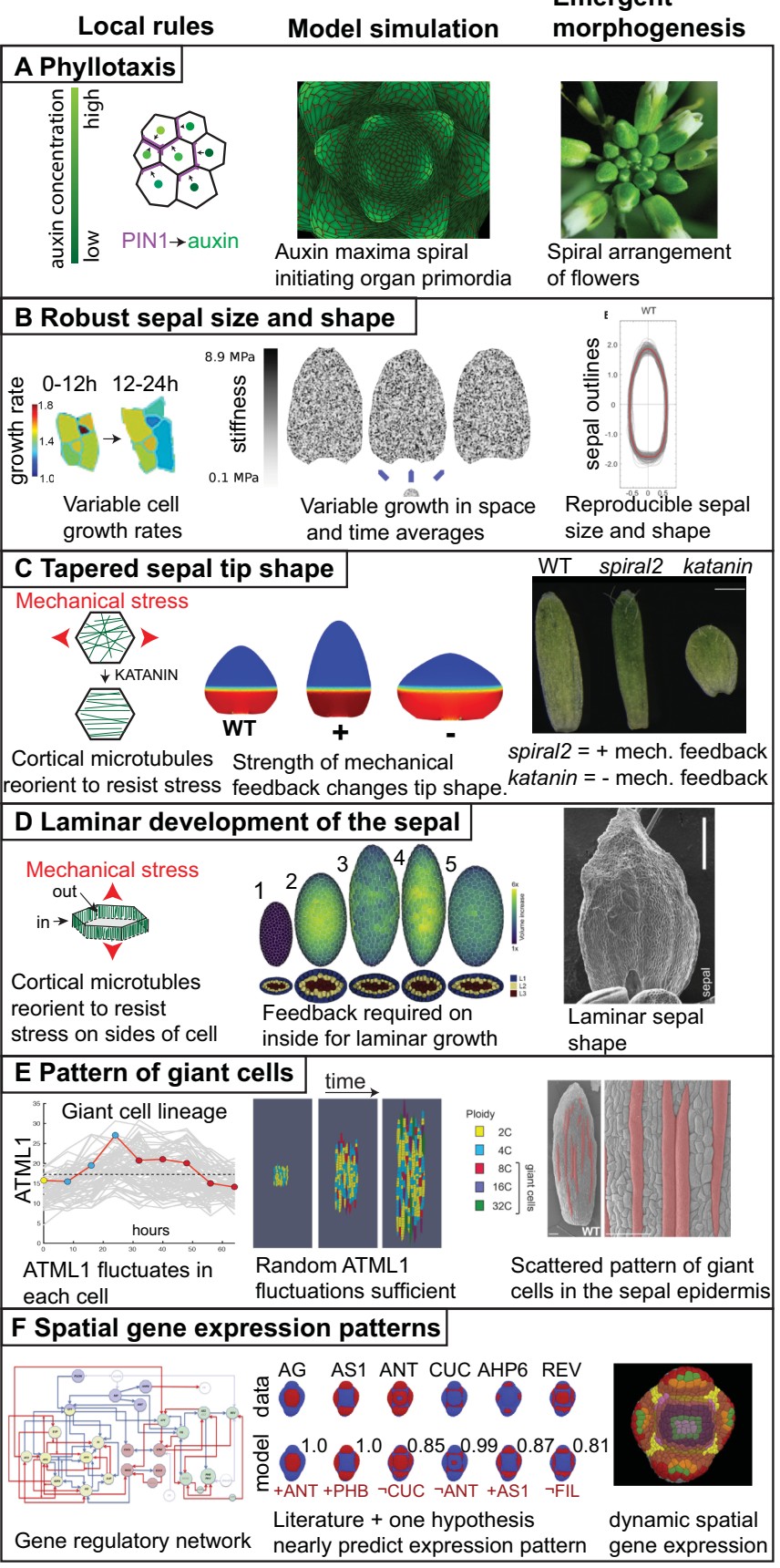

**Local rules**  **Model simulation**  **Emergent morphogenesis**

**A Phyllotaxis**

auxin concentration (low–high)

PIN1 → auxin

Auxin maxima spiral initiating organ primordia

Spiral arrangement of flowers

**B Robust sepal size and shape**

growth rate 0-12h 12-24h (1.0, 1.4, 1.8)

stiffness (8.9 MPa – 0.1 MPa)

WT — sepal outlines

Variable cell growth rates

Variable growth in space and time averages

Reproducible sepal size and shape

**C Tapered sepal tip shape**

Mechanical stress

KATANIN

WT + −

WT  *spiral2*  *katanin*

Cortical microtubules reorient to resist stress

Strength of mechanical feedback changes tip shape.

*spiral2* = + mech. feedback
*katanin* = − mech. feedback

**D Laminar development of the sepal**

Mechanical stress
out
in

1 2 3 4 5

Volume increase (1x–6x)
L1 L2 L3

sepal

Cortical microtubules reorient to resist stress on sides of cell

Feedback required on inside for laminar growth

Laminar sepal shape

**E Pattern of giant cells**

time

Giant cell lineage

ATML1 / hours

Ploidy
2C
4C
8C
16C
32C

giant cells

WT

ATML1 fluctuates in each cell

Random ATML1 fluctuations sufficient

Scattered pattern of giant cells in the sepal epidermis

**F Spatial gene expression patterns**

data
model

AG  AS1  ANT  CUC  AHP6  REV

1.0  1.0  0.85  0.99  0.87  0.81

+ANT  +PHB  ¬CUC  ¬ANT  +AS1  ¬FIL

Gene regulatory network

Literature + one hypothesis nearly predict expression pattern

dynamic spatial gene expression

In my career as a plant biologist, I have focused on developing *Arabidopsis* sepals as a system for elucidating emergent properties of morphogenesis. In this review, first I describe the advantages of the sepal system, explaining why I chose this system. Then, I look at sepals from the perspective of a modeller embarking on simulating sepal development and describe the quantitative fundamentals of this system. Next, I present five case studies in which we have learnt about the emergent processes driving morphogenesis from sepals. Finally, I conclude with some of the big challenges for the future.

## 2. Sepal definition

Sepals are the outermost organs of the flower (Figure 2a). Sepals typically form in a whorl surrounding the petals. The set of sepals make up the calyx. The word sepal originates from New Latin *sepalum*, which was coined by Nöel Martin Joseph de Necker in 1790 and derived from Greek σκέπη, meaning covering (de Necker, 1790; Oxford English Dictionary Online, 2021). Typically, sepals are green photosynthetic organs that enclose and protect the developing flower bud before it blooms (Figure 2a,k). However, a wide range of sepal coloration and morphology is present across the angiosperms, and in some cases the sepals are showy petaloid organs. When the sepals and petals are similar in a flower these organs are called tepals, particularly in flowers of 'basal' angiosperms such as *Amborella trichopoda* and monocots such as lilies (Buzgo et al., 2004). In monocots of the grass family, there is debate about the homology of floral organs, but the outermost floral organs, the lemma and palea, may be equivalent to sepals (Ambrose et al., 2000). Eudicot flowers commonly have four sepals (e.g., *Arabidopsis*) or five sepals (e.g., petunia) while monocots have three or multiples of three sepals.

Organ identity regulates the final form of the organ; sepals have very different shapes and sizes from leaves, petals, stamens and carpels. In *Arabidopsis*, sepal identity is specified by the A-function genes *APETALA1* (*AP1*) and *APETALA2* (*AP2*) in the classic ABC model of floral organ identity (Bowman et al., 1989; 1991; Chen, 2004; Coen & Meyerowitz, 1991; Irish & Sussex, 1990; Kaufmann et al., 2010; Mandel et al., 1992; Monniaux et al., 2018). In *ap1* mutants, the sepal is converted into a bract, which has a larger size and more leaf-like shape (Mandel et al., 1992). In strong *ap2* mutants, the sepal is converted into an unfused carpel, whereas in weak *ap2* mutants, it is converted into a larger leaf (Bowman et al., 1989, 1991). The molecular underpinning of the ABC model largely consists of MADS box transcription factors binding DNA in quartet complexes that also include SEPALATA transcription factors (Ditta et al., 2004). A function involves not only specifying sepal and petal identity, but also repressing C function, limiting it to the third and fourth whorls. Genome wide targets of AP1 and AP2 have been identified and include many transcription factors (Kaufmann et al., 2010; Yant et al., 2010), but how these genes control the growth rates and polarity fields needed to give rise to organ morphogenesis remains unclear and is one of the big challenges for the future (Green et al., 2010; Kennaway et al., 2011; Kuchen et al., 2012; Rebocho et al., 2017; Sauret-Güeto et al., 2013; Thomson et al., 2017).

Outside of *Arabidopsis*, A function has remained mysterious because in many other species *AP1* orthologs function primarily in floral meristem (FM) identity and do not control both sepal and petal identity (Litt, 2007). Likewise *AP2* orthologs show little evidence of repressing C function (Litt, 2007). However, Morel et al. have found that genes encoding A function are more diverse in petunia than *Arabidopsis* (Morel et al., 2017). The EuAP2 transcrip-

tion factors, REPRESSOR OF B-FUNCTION (ROB), do have roles in sepal and petal development of petunia, but do not repress C-function. In contrast, a TOE-type AP2 transcription factor, BLIND ENHANCER (BEN), and the microRNA *BLIND* repress C function in the first and second whorls of petunia (Morel et al., 2017), carrying out the major A function. Thus, we need to be cautious in our extrapolation from *Arabidopsis* development to other species, but often the same underlying principles can be found even when the genes are not the same (Koonin et al., 1996; Striedter, 2019).

## 3. Advantages of *Arabidopsis* sepals as a model system for morphogenesis

The *Arabidopsis thaliana* sepal is an advantageous system for studying morphogenesis because of its simplicity, accessibility and reproducibility of morphogenesis. The simplicity of the final sepal shape entices research to focus on fundamental questions of morphogenesis, such as the attainment of size and shape. As the outermost floral organ, the sepal is accessible for imaging and manipulation. Although older flowers overlie younger ones, they can be dissected from the inflorescence to reveal the sepals of interest from their initiation on the FM (Zhu et al., 2020). The entire morphogenesis of the living sepal can be imaged on the microscope (Zhu et al., 2020). Crucially, the final size of the *Arabidopsis* sepal is only about 1 mm$^2$, so that even at its mature size, it can be captured with cellular resolution within a few images stitched together. In contrast, *Arabidopsis* leaves become so large that imaging the entire leaf with cellular resolution becomes exceedingly challenging to nearly impossible. Robustness, or reproducibility of outcome in the face of environmental and other perturbations, is essential for sepal function, and makes morphogenesis predictable. The presence of four sepals on each flower and over 100 flowers on each plant allows enough statistical power to assess robustness of sepal morphogenesis using a single plant (i.e., single genetic condition and environment). In wild type, the size of the sepals is robust for the four sepals in a flower, the sepals of different flowers on the same plant (especially for flowers 10–25 on the main branch), and for flowers on different plants (Hong et al., 2016; Zhu et al., 2020). In addition, the tremendous resources of *Arabidopsis* such as sophisticated functional genomics, mutant collections, CRISPR, easy generation of transgenics, and so forth, which have been built up over decades of concerted research can be harnessed to understand morphogenesis (Provart et al., 2015).

In this article, I focus on *Arabidopsis* sepals because they have become a useful experimental system as the basis for understanding morphogenesis using computational biology and modelling. Hopefully, what we learn in *Arabidopsis* will extend not just to sepals in other flowers, but also more generally to the morphogenesis of lateral organs in plants. The morphogenesis of leaves and sepals is similar (see Section 6), meaning that we can learn fundamental principles of lateral organ morphogenesis that are likely to be conserved. In fact, research on floral organ identity has shown that sepals can be converted into leaves through the mutation of organ identity genes (Bowman et al., 1991). Conversely overexpression of floral organ identity genes is sufficient to convert leaves into floral organs (Pelaz et al., 2001). This suggests that there is an underlying basal organ program as Goethe suggested many years ago (Goethe, 1790). Supporting this view, only 13 genes were found to be specifically expressed in sepals based on transcriptomic analysis of floral organs; most genes expressed in sepals are also expressed in other organs (Wellmer et al., 2004).

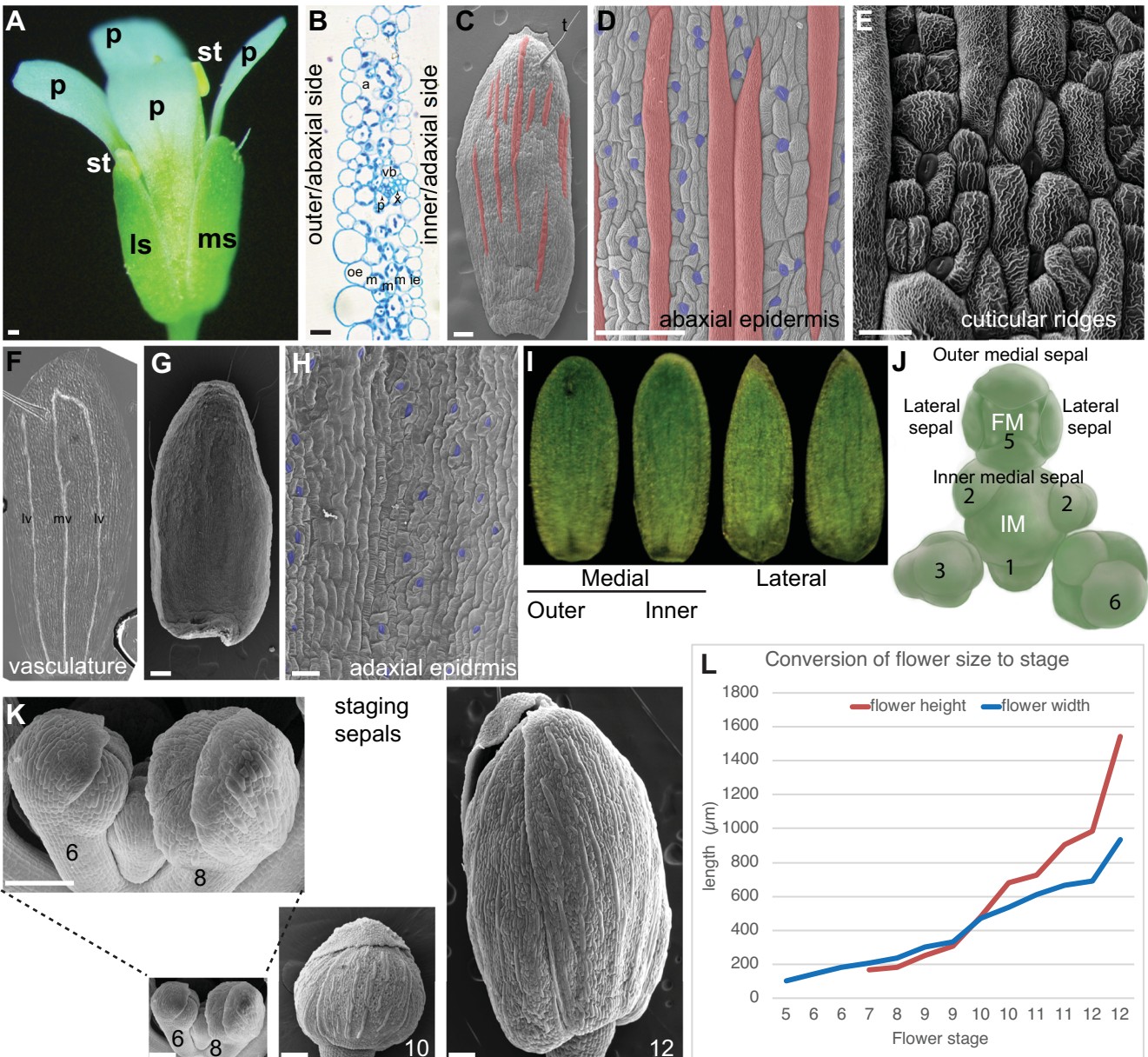

**Fig. 2.** *Arabidopsis* sepal morphology. (a) Photograph of a mature *Arabidopsis* flower (stage 14). A medial (ms) and lateral (ls) sepal are visible, as well as all four petals (p) and two of the stamens (st). Scale bar: 100 μm. (b) Cross section of part of a sepal. The outer or abaxial side faces left and the inner or adaxial side faces right. The five layers of cells from outer epidermis (oe), three layers of mesophyll (m) and the inner epidermis (ie) are visible. Air spaces (a) and one vascular bundle with phloem (p) and xylem (x) are visible. Scale bar: 10 μm. Image courtesy of Lilan Hong. (c) Scanning electron micrograph (SEM) of an *Arabidopsis* Colombia-0 (Col-0) sepal showing the abaxial epidermis with giant cells (false coloured red in Photoshop). One trichome (t) is present at the tip of the sepal. Scale bar: 100 μm. Image reproduced under the CCBY 4.0 licence from Meyer et al. (2017). (d) SEM the abaxial (outer) sepal epidermis (Col-0) with giant cells (false coloured red in Photoshop) interspersed between smaller pavement cells (not coloured) and guard cells (false coloured blue). Scale bar: 100 μm. Image reproduced and modified under the CCBY 4.0 licence from Meyer et al. (2017). (e) SEM showing the culticular ridges (white wavy lines) decorating the abaxial sepal epidermal cells in an *Arabidopsis* Landsberg *erecta* (L*er*) sepal. Note that guard cells do not form ridges. Scale bar: 20 μm. Image courtesy of Clint Ko and Adrienne Roeder. (f) Stage 12 sepal that has been cleared to reveal the vasculature: midvein (mv) and lateral veins (lv). Image courtesy of Frances Clark. (g) SEM of the adaxial (inner) side of the sepal (L*er*) showing the curved shape. (h) SEM of the adaxial (inner) sepal epidermis (L*er*) with guard cells (false coloured blue). Note the absence of giant cells. Scale Bar: 30 μm. (i) Images of outer medial, inner medial, and lateral sepals from one *Arabidopsis* flower Col-0 accession. Images courtesy of Lilan Hong. (j) Drawing of the *Arabidopsis* inflorescence meristem (IM) with a flower at the top, in which the position of the four initiating sepal primordia relative to the IM and floral meristem (FM) is indicated. Image reprinted from Zhu et al. (2020). (k) SEMs of *Arabidopsis* Col-0 sepal morphogenesis at developmental stages 6, 8, 10 and 12 from when the sepals first close around the FM (stage 6) to the last point the bud remains closed before blooming (stage 12). Images are at the same magnification to display the growth of the sepals. Inset is a magnified view of stages 6 and 8. (l) *Arabidopsis* Col-0 flower height and width at each stage of flower development. See also Table 1.

## 4. The outcome of morphogenesis: The mature *Arabidopsis* sepal architecture

To understand the emergent process of sepal morphogenesis, we first have to consider its outcome: in other words, the morphology of the mature sepal. To model sepal morphogenesis, we need a quantitative and detailed description to determine whether our models are sufficient to simulate sepal development or not.

The four sepals of each flower have been given names to distinguish them based on their positions relative to the inflorescence meristem (IM) from which the flower initiated (Zhu et al., 2020).

The outer medial (abaxial) sepal is farthest from the IM, while the inner medial (adaxial) sepal is closest to the IM (Figure 2i,j). The sepals on the sides of the flower are called the lateral sepals. The medial sepals both have rounded tips, whereas the lateral sepals have pointed tips. The outer sepal tip is the outermost, overlapping the other sepal tips. The four sepals are nearly the same size (Figure 2i).

The mature *Arabidopsis* sepal is a slightly curved, green photosynthetic organ, 1.09 mm$^2$ (±0.15 mm$^2$ SD) in area (Figure 2a,i; Zhu et al., 2020). The sepal is approximately 5 cell layers thick, consisting of an abaxial (outer) epidermis, three layers of mesophyll cells, and an adaxial (inner) epidermis, facing the petal (Figure 2b). The abaxial epidermis consists of 1,590 cells (±320 cells SD) divided between pavement cells, guard cells surrounding stomatal pores, and possibly a few trichomes (hair cells; Figure 2c; Roeder et al., 2010). The sepal pavement cells have relatively elongated rectangular shapes, unlike leaf pavement cells which have jigsaw puzzle shapes (Bowman, 1994; Sapala et al., 2018). Sepal pavement cells form in a large variety of sizes, which have been divided into two different cell types: small cells and giant cells (Figure 2d; see Section 7.4; Roeder et al., 2010). Giant cells are on average 360 μm (±150 μm SD) long and form through endoreplication (replication of DNA without division) to about 16C. The presence of giant cells is used as a marker for sepal organ identity, distinguishing them from other floral organs (Pelaz et al., 2000). Within the sepal, giant cells are only present in the abaxial epidermis, and are not found in the mesophyll cells or the adaxial epidermis (Roeder et al., 2010).

The abaxial sepal pavement cells are decorated with cuticular nanoridges (Figure 2e). The structure of the nanoridges is wavy and convoluted, and it has been speculated that the pattern forms via mechanical buckling (Antoniou Kourounioti et al., 2012; Hong et al., 2017; Smyth, 2017). Nanoridges, being made of cutin, are lost in mutants that disrupt cutin biosynthesis and transport (Hong et al., 2017; Li-Beisson et al., 2009; Panikashvili et al., 2009; Shi et al., 2011).

Guard cells are 29% (±3% SD) of cells in the abaxial sepal epidermis (Roeder et al., 2010). Guard cells are generally the smallest of the cells in the sepal epidermis. Stomatal patterning ensures at least one cell spacing between stomata, similar to leaves (Figure 2d) (Herrmann & Torii, 2020; Lee & Bergmann, 2019). Guard cells contain chloroplasts, while the other epidermal cells do not.

Sepal trichomes have a single branch and form through endoreplication to about 16C (Perazza et al., 1999; Figure 2c). More trichomes are found on the sepals of the first few flowers formed when the plant transitions to flowering and bolts than on subsequent flowers. Many sepals formed later do not have any trichomes.

Mesophyll cells contain chloroplasts, making the sepals green and photosynthetic. The mesophyll cells are separated by air spaces, particularly under the stomata (Figure 2b). Mesophyll cells are excluded from the sepal margins, which consist of the two epidermal layers appressed to each other. Thus, the margins are white, due to the absence of chloroplasts in epidermal cells. The sepal margins are quite distinct from the leaf margins and lack the characteristic highly endoreduplicated cells of leaf margins.

The vasculature of the sepal is relatively simple, typically comprising a midvein in the centre of the sepal and two looping lateral veins nearer the margins (Figure 2f). These veins usually, but not always connect. The vein architecture is somewhat variable and additional branch veins may be present. The vascular tissue passes through the mesophyll layers of the sepal, closer to the adaxial side than the abaxial side (Figure 2b).

The adaxial epidermis contains 2,820 ± 440 (SD) cells composed of pavement cells and guard cells (Figure 2g,h; Roeder et al., 2010). The pavement cells are more uniform in size. No trichomes, giant cells, or cuticular nanoridges are present on the inner epidermis.

## 5. *Arabidopsis* sepal development and staging

Sepal development is staged according to stages of *Arabidopsis* flower development that were defined by Smyth et al. (1990). The sepals are the first organs to initiate from the FM, being clearly visible at stage 3 (Figure 2j). The sepal primordia grow rapidly and cover the FM by stage 6 (Figure 2j,k). While the flower develops (stages 7–12), the four sepals curve inward to enclose and protect the internal developing organs: four petals, six stamens (male reproductive organs), and two fused carpels (female reproductive organs; Figure 2a,k). The four sepals must maintain the same approximate size throughout their growth to ensure the closure of the flower bud (Hong et al., 2016; Zhu et al., 2020). In addition, their growth must accommodate that of the internal organs. The sepals straighten up and are pushed open by the petals and internal organs when the flower blooms (anthesis; stage 13). The sepals remain upright surrounding the base of the flower. After fertilisation of the flower, at stage 16, the sepals senesce, wither, and fall from the flower as the fruit starts to develop.

It is often essential to be able to conduct experiments on sepals of the same developmental stage and understand the stage of a sepal in the developmental progression. However, many of the flower staging landmarks described by Smyth et al. occur in internal organs that are not visible through the sepals of intact flowers (Smyth et al., 1990). Moreover, the Smyth et al description describes the Landsberg *erecta* (L*er*) ecotype, but the field has since largely switched to using Columbia-0 (Col-0) as the standard. Therefore, I have established a graph converting the Smyth et al flower stage to Col-0 flower width (generally the least altered measurement in mutants) and flower height (Table 1 and Figure 2l). Using this table, the Smyth et al flower stage can be approximated from measurements of closed floral buds.

**Table 1.** Flower stage versus size

| Stage | Flower height (μm) | Flower width (μm) |
|---|---|---|
| 5 | | 103 |
| 6 | | 140 |
| 6 | | 180 |
| 7 | 166 | 206 |
| 8 | 183 | 238 |
| 9 | 254 | 301 |
| 9 | 306 | 330 |
| 10 | 480 | 469 |
| 10 | 682 | 537 |
| 11 | 726 | 613 |
| 11 | 903 | 663 |
| 12 | 983 | 692 |
| 12 | 1,541 | 937 |

*Note:* Since the flowers grow during these stages, for some stages multiple measurements are giving to provide a range. Height is measured along the proximal distal axis of the bud (i.e., pedicel to tip) and width on the medial lateral axis (across the widest part of the bud).

## 6. Growth pattern of sepals

How do the patterns of cellular growth give rise to the final sepal form? Similar to leaves in *Arabidopsis* and many other species (Gupta & Nath, 2015), *Arabidopsis* sepals exhibit a basipetal pattern of growth (Figure 3; Hervieux et al., 2016). Live imaging has shown that from stage 4 through stage 6, after the sepal first initiates, the cells of the sepal grow at the very rapid rate of 200–400% increase in cell area in 24 hr, particularly near the tip (Figure 3a; Hervieux et al., 2016). At stage 7, the overall growth slows to the still relatively fast rate of 100–200% increase in cell area in 24 hr throughout the sepal and with the fastest growth (200%) concentrated along the margins (Figure 3a). Starting at stage 8, growth dramatically slows at the tip of the sepal to 0–50% increase in cell area, while a zone of fast growth (100–200%) is maintained throughout the middle and base of the organ (Figure 3a). As the sepal continues to grow at stage 9, the region of slow growth at the tip expands, while the fast growth zone is maintained in approximately the same size at the base of the sepal (Figure 3a). At stage 10, growth slows to 0–50% increase in cell area throughout much of the sepal (Figure 3a), although considerable slow expansion continues to occur through stage 12. At early stages 4–6, the growth of the sepal is highly anisotropic (growing more in the tip to base orientation than from side to side), but becomes more isotropic (growing equally in all directions) at later stages 7–10 (Figure 3b; Hervieux et al., 2016). Throughout this process, cell division primarily occurs in the region of fast growth, whereas slow cell expansion without division occurs in the region of slow growth (Figure 3c; Hervieux et al., 2016; Roeder et al., 2010). Endoreduplication typically occurs together with cell division in the fast growth zone, where neighbouring cells either are mitotically dividing or endoreduplicating (Roeder et al., 2010). Cuticular nanoridges form on the cells undergoing slow cell expansion and are absent from the fast growth and division zone (Hong et al., 2017). Although this growth pattern is often described as a basipetal wave of growth slowing first at the tip and progressing downward, an equally valid way of describing it is that the sepal produces cells from the fast growth zone at the base. As more cells are produced, they are pushed upward, and proceed from the fast growth zone into the cell expansion zone as they mature (Roeder et al., 2010).

Within this overarching growth pattern of the whole sepal, the growth rate of individual cells is highly heterogeneous (Tauriello et al., 2015). Note that this heterogeneity is not shown in Figure 3, because the growth of neighbouring cells has been smoothed to reveal the overarching patterns among the noise. The source of the heterogeneity is not entirely clear, but some interesting quantitative patterns have been detected. One factor contributing to the heterogeneity in growth is asynchrony of the cells along their growth curves. The size of each cell clone (the progeny of a single progenitor cell) follows a sigmoidal or S-shaped curve, with the growth rate accelerating, reaching a maximum, decelerating, and eventually stopping. Interestingly all cell lineages reach the same maximum relative growth rate (growth rate divided by size), but they reach this maximum at different times during the development of the sepal (Tauriello et al., 2015). Thus, when observing a single time point, the cells appear to be growing at different rates because they are on different phases of their individual growth curves.

A second factor contributing to cellular heterogeneity in sepals is variability in the size of a mother cell when it divides (Roeder et al., 2010; Schiessl et al., 2012). In the IM, there is a tight correlation between the volume of a mother cell and its entry into the cell cycle, specifically the initiation of DNA replication at the transition between G1-S phases (Jones et al., 2017; Schiessl et al., 2012). In contrast, this correlation is lost in sepal primordia, and cells initiate DNA replication at a wide variety of volumes (Schiessl et al., 2012). The JAGGED (JAG) transcription factor, which is expressed in sepals but not the meristem, causes the relaxation of cell size thresholds for cell division in the sepal (Schiessl et al., 2012; 2014).

The variability in cell size after division also contributes to the growth rate heterogeneity. Clones derived from small progenitor cells grow faster to catch up with their larger neighbours, leading to more uniform clone sizes for the first 24–60 hr (Tsugawa et al., 2017). However, after 24–60 hr, such compensatory growth disappears, perhaps because the daughter cells within the clone are now undergoing their own processes to maintain size uniformity (Tsugawa et al., 2017). Meristem cells also exhibit compensatory growth behaviour in which smaller daughter cells grow more quickly to catch up with their larger sisters (Willis et al., 2016).

Surprisingly, this cellular heterogeneity in growth rate appears to have little effect on the overall growth of the organ. Scientists have simulated the growth of the sepal starting from the initial cells at the first time point of a live imaging series (Tauriello et al., 2015). They have simulated what would happen if the cells grew at a uniform growth rate instead of the heterogeneous rates observed. When they compared the size and shape of the simulated clones with the actual clones from the live imaging series, the two matched surprisingly well (Tauriello et al., 2015). This result suggests that the variability averages out in time, such that the cumulative growth is nearly equivalent to a uniform growth rate (see Section 7.1). Further research revealed that cell growth hetereogenetiy is actually required for spatiotemporal averaging to occur and consequently cellular heterogeneity promotes robustness of sepal size and shape (Hong et al., 2016).

Many questions remain about how this growth pattern gives rise to the overall sepal size and shape. Why do the cells grow heterogeneously? Is there some advantage over uniform growth? How does robust sepal size and shape emerge from the variable growth and division of the constituent cells (see Section 7.1)? How is the zone of fast growth at the base of the sepal established and what controls its size? This growth pattern is consistent with the hypothesis that a growth promoting morphogen diffuses from the base of the organ, which has been proposed and modelled in leaves (De Borger et al., 2015; Fox et al., 2018; Kazama et al., 2010; Kuylen et al., 2017). What triggers the slowing of growth throughout the sepal at stage 10 and the termination of the fast growth zone at the base? Does the sepal measure its size in some way to make this transition?

## 7. Case studies of emergent processes in sepal morphogenesis

Based on the above description of sepal morphogenesis, I will now consider a few examples of emergent phenomena that have been uncovered and found to contribute to sepal morphogenesis (Figure 1). In each example, I will focus on how the simulation of simple cellular-scale local rules gives rise to emergent behaviour at the sepal scale. While these models do not capture the entire complicated, nuanced, underlying set of biological mechanisms, they are invaluable in their elucidation of the logical processes behind these mechanisms.

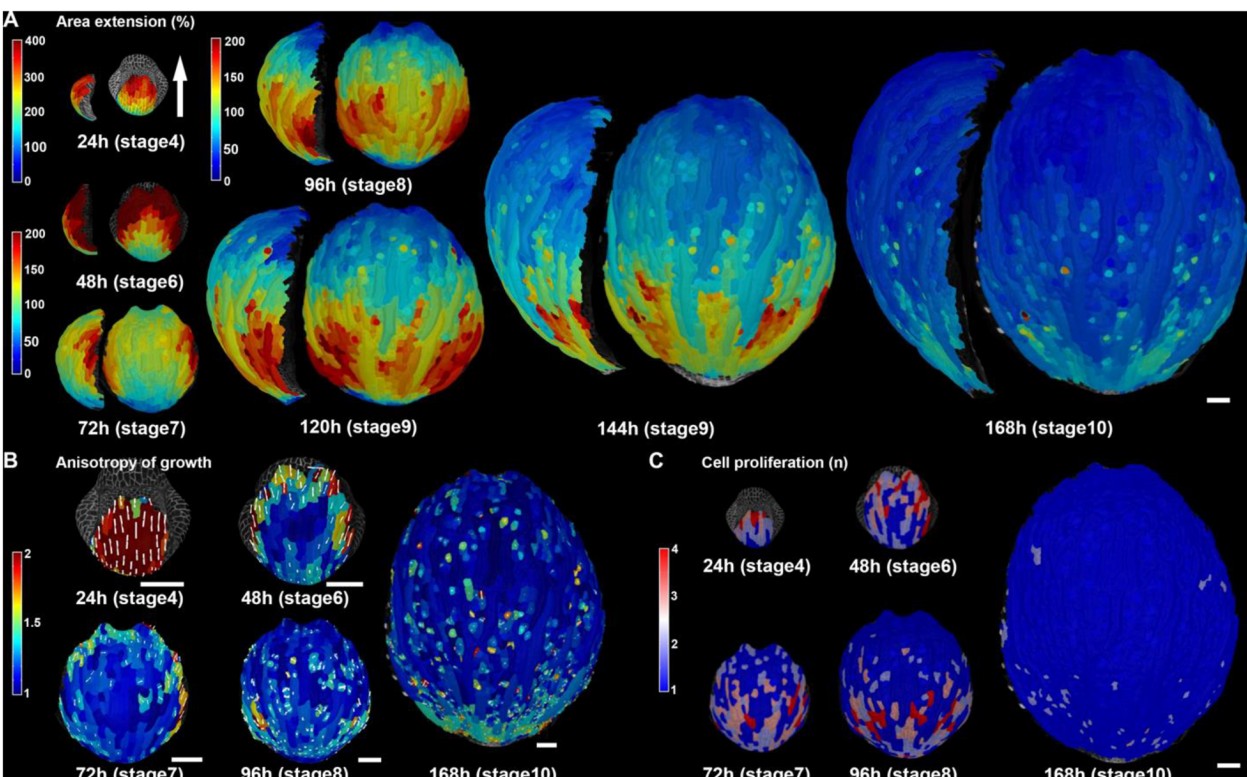

**Fig. 3.** *Arabidopsis* sepal growth. (a) Heat map of growth rate displayed as percent area extension over consecutive 24 hr intervals with associated stages. The growth rates are averaged to smooth the variability in cell growth rates and reveal the overarching basipetal growth pattern. The sepals are displayed at the same magnification with a side view to the left of front view. Note the heat map scale for the 24 hr stage is different from the later stages. (b) Heat map of the anisotropy of growth over consecutive 24 hr intervals. The maximal growth direction is indicated by a white line if anisotropy is >20%. The anisotropy is calculated as the growth in the maximal direction divided by the minimal direction. (c) Heat map of cell proliferation over consecutive 24 hr intervals. The heat map displays the number of daughter cells descending from one mother cell. In other words, if there is one daughter cell, no division took place. Note that division tends to occur in fast-growing regions. Scale bars: 50 μm. This figure in its entirety was reprinted from Hervieux et al. (2016) with permission from Elsevier.

## 7.1. Emergence example 1: Robustness of sepal size and shape

*Emergent phenomenon*: Sepals form robustly with the same size and shape despite heterogeneity in the growth rates and division patterns of their constituent cells (Figure 1b, right).

*Local rule*: Each cell varies its growth rate in space and time (Figure 1b, left).

*Modelling*: Simulation of 'cells' which randomly choose their growth rates at each time interval of the model produces robust sepals (Figure 1b, middle).

As mentioned above, one of the big mysteries in sepal morphogenesis is how heterogeneous growth, division and size of sepal cells give rise to highly reproducible sepal organ size and shape (Figure 1b, right). Growth of plant cells is controlled by the mechanical properties of their cell walls. The stiffer the cell wall, the less it can expand. Parallel to the heterogeneity in growth rate (Figure 1b, left), atomic force microscopy (AFM) has been used to show that there is considerable heterogeneity in cell wall stiffness, even within a single sepal cell (Hong et al., 2016).

Mathilde Dumond and Arezki Boudaoud tackled this problem through modelling (Hong et al., 2016). First, they developed a finite element model of a single sheet of tissue, with uniform stiffness that would expand into a sepal shape when simulated. Then they altered the stiffness of the initial model to reflect the spatial variability of cell wall stiffness observed in the AFM data. When growth was simulated, this model failed spectacularly. In each case the stiff spots failed to grow and the soft spots massively overextended,

creating bizarre forms. However, in the real sepal, the growth rate of a cell varies in time, which inspired them to allow the stiffness of the model to vary in time as well as space (Hong et al., 2016). At each time point in the simulation, each triangle of the model material randomly chose its stiffness from a probability distribution based on the stiffness distribution observed in the AFM data. Remarkably, this model produced robust sepal shapes in each simulation run (Figure 1b, middle). In the model, the varying stiffness averaged out over time, in a process called spatiotemporal averaging.

The spatiotemporal averaging that occurs in the model was subsequently verified to occur in living sepals through the analysis of live imaging data (Hong et al., 2016). Note that in this model the only communication occurring between 'cells' is mechanical, when they pull or push on one another. Thus robust sepal size and shape emerge from the simple local rule that each cell wall varies its stiffness and consequently its growth rate in time (Hong et al., 2016).

In contrast to pavement cells, which undergo spatiotemporal averaging of their variable growth rates, trichome cells exhibit sustained rapid growth, extending out of the surface of the sepal. The base of the trichome is also rapidly expanding and pushes on its neighbours. Wildtype sepal shape is robust to variation in trichome number (Hervieux et al., 2017). The local cell behaviour that is responsible for buffering the effects of trichome growth is a mechanical feedback loop. The fast-growing trichome cell pushes on its neighbours creating mechanical stress (Hervieux et al., 2017). The cortical microtubules in the neighbouring cells reorient to

reinforce against stress (Hamant et al., 2008; Sampathkumar et al., 2014), creating a concentric ring around the trichome and constraining its growth and buffering its effect on the sepal (Hervieux et al., 2017). Simulations of models with this feedback loop show it is sufficient to buffer the effects of trichomes on organ shape. In contrast, simulating models without the feedback loop show that trichomes are sufficient to generate variability in sepal organ shape. Specifically, increased numbers of trichomes correlate with increasing the width of the sepal. This model prediction was tested by examining the effect of trichome number on the robustness of sepal shape in *katanin* mutants in which the mechanical feedback is inhibited. *KATANIN* encodes a microtubule severing protein, and severing promotes microtubule dynamics allowing microtubules to reorient in response to mechanical stress (Bichet et al., 2001; Burk & Ye, 2002; Hamant et al., 2008). This reorientation is inhibited in *katanin* mutants (Hervieux et al., 2016; Sampathkumar et al., 2014; Uyttewaal et al., 2012). Consequently, in *katanin* mutants, the shape of the sepals varies with trichome number; sepals with more trichomes are wider than those with fewer (Hervieux et al., 2017). Thus, spatiotemporal averaging and mechanical buffering of fast-growing cells are two mechanisms producing the emergent phenomenon of shape robustness. There are probably many more mechanisms remaining to be discovered.

Timing often appears to be crucial in emergent phenomena. In addition to the spatiotemporal averaging of heterogeneity in cellular growth, the synchronous initiation of sepal development on the FM (Zhu et al., 2020) and the synchronous termination of sepal growth (Hong et al., 2016) are both critical for robustness in sepal size. The proper timing of sepal initiation requires the formation of focused auxin and cytokinin signalling zones at the position of each sepal before they emerge (Zhu et al., 2020). At the end of sepal development, reactive oxygen species (ROS) are a major signal for maturation and the termination of growth (Hong et al., 2016). However, what causes the accumulation of ROS at the correct time to synchronise maturity remains unknown. Further insight into emergence in morphogenesis will require understanding how the timing of developmental events is regulated. Is timing itself an emergent property that arises from network motifs and communication between cells during morphogenesis?

### 7.2. Emergence example 2: Sepal tip shape emerging from mechanical stress

*Emergent phenomenon*: The tip of the sepal is tapered (Figure 1c, right).

*Local rule*: Cortical microtubules in each cell reorient in response to mechanical stress (Figure 1c, left). These microtubules guide the deposition of cellulose, reinforcing the cell wall in that direction. The reinforced cell wall changes the anisotropy of the cell's growth and orients growth perpendicular to the main stress.

*Global growth pattern*: At a larger scale, the basipetal growth pattern of the sepal in which a slow growth zone at the sepal tip is connected with a fast growth zone at the base, creates tensile mechanical stress, to which microtubules in the cells respond.

*Modelling*: Modelling the cell's mechanical feedback loop and a basipetal growth gradient was sufficient to generate a range of tip shapes (Figure 1c, middle).

Another fundamental question is how the shape of the sepal is generated, in this case the tapered tip shape (Figure 1c, right). As mentioned above in Section 6, the sepal growth slows in a basipetal gradient descending from tip to base. This growth pattern creates a mechanical conflict between the slow-growing cells in the tip and the fast-growing cells just beneath them (Hervieux et al., 2016). At the local level, cells respond to mechanical stress by reorienting their microtubules in a pattern to resist the stress (Figure 1c, left), as mentioned above with regard to robustness of sepal shape to trichome number (Hervieux et al., 2017). Microtubules form tracks along which cellulose synthase complexes move, attached by cellulose synthase interactive (CSI) proteins (Bringmann et al., 2012; Li et al., 2012; Paredez et al., 2006). Thus the cortical microtubule array in turn orients the newly synthesised cellulose microfibrils in the cell wall, reinforcing the cell wall against stress (Hamant et al., 2008; Hervieux et al., 2016; Sampathkumar et al., 2014). The mechanisms through which mechanical stress is sensed and transmitted to microtubule orientation are not well understood. Cellulose microfibrils are not extensible, so the newly deposited cellulose microfibrils alter the direction and anisotropy of growth, consequently changing the cell shape. Thus, the current mechanical stresses, alter the growth direction and change the cell shape, which generates a new pattern of mechanical stresses, making a feedback loop.

Simulating models of the basipetal growth gradient in addition to this mechanical feedback loop, creates a range of tapered tip shapes (Figure 1c, middle). In these models, decreasing the mechanical feedback produces a rounder tip, whereas increasing mechanical feedback increases tapering producing a more triangular tip. These model predictions were verified experimentally (Figure 1c, right). The response of microtubules to mechanical stress is dampened in the *katanin* mutant, and the sepal tip is broader and more rounded, as is the whole sepal, supporting the model. In contrast, the response of microtubules to mechanical stress is enhanced in *spiral2* mutants and the sepal tip is longer and more tapered, as predicted (Hervieux et al., 2016). *SPIRAL2* encodes a microtubule associated protein that regulates cortical microtubule dynamics, particularly binding to and stabilising the minus end (Fan et al., 2018; Nakamura et al., 2018; Wightman et al., 2013). It has been experimentally shown that microtubules reorient faster after mechanical perturbations in *spiral2* mutants than wild type, indicating they are more responsive to mechanical stress in the mutant (Hervieux et al., 2016). The feedback loop between mechanical stress, microtubules and growth orientation appears to be a core component of many emergent processes in organ morphogenesis, as we will see again in laminar development of the sepal (Section 7.3).

### 7.3. Emergence example 3: Laminar development of the sepal

*Emergent phenomenon*: The sepal forms an extended relatively flat structure (Figure 1d, right).

*Local rule*: Only the cortical microtubules on the inner walls of the epidermal cells and the inner cells of the sepal reorient in response to mechanical stress (Figure 1d, left).

*Modelling*: A three-dimensional mechanical model in which the mechanical response of the microtubules occurs in the inner cell walls, and not the outer cell walls, grows to create a flat laminar structure (Figure 1d, middle).

How does a laminar plant organ such as a flat leaf blade or sepal form during morphogenesis (Figure 1d, right)? This question fascinated Zhao et al., who examined the process in sepals and leaves (Zhao et al., 2020). They noticed that the cortical microtubules were better aligned with the direction of mechanical stress on the sides of sepal and leaf cells than on the outer surface. This observation suggested to them the local rule that microtubules on the inner cell

walls, but not the outermost cell wall might be orienting in response to mechanical stress (Figure 1d, left).

Zhao et al. modelled the possibilities with a three-dimensional cellular mechanical model (Figure 1d, middle; Zhao et al., 2020). Based on the structure of leaf and sepal primordia, they initiated the model with an oblong shape. If they simulated the model with no feedback between mechanical stress pattern and microtubule orientation restricting growth, then the sepal bulged towards a rounded shape (simulation 2). Likewise, simulating the model with mechanical feedback only in the outer cell wall caused the sepal to bulge and not make a flattened lamina (simulation 4). Simulating mechanical response throughout all cell walls did produce flattening of the lamina, but growth was disproportionately concentrated in lengthening the organ and did not match in vivo data (simulation 3). In contrast, if they simulated the model with mechanical stress orienting microtubules only in the inner cell walls, but not the outer cell walls, they recapitulated laminar growth of a flat organ (simulation 5). Their model likewise showed that the initial shape of the organ was critical; a spherical primordium could not form a lamina (Zhao et al., 2020). Thus, the combination of highly localised mechanical feedbacks within the cell together with the initial oblong primordium shape is what generates the emergence of a flat laminar organ (Figure 1d).

### 7.4. Emergence example 4: Cell size patterning

*Emergent phenomenon*: Giant cells form scattered among a range of smaller cells in the sepal epidermis (Figure 1e, right).

*Local rule*: The concentration of the ATML1 transcription factor fluctuates in each cell (Figure 1e, left). If ATML1 reaches a high concentration during G2 phase of the cell cycle, the cell is likely to endoreduplicate. Endoreduplicating cells are terminally differentiated and generally cannot divide.

*Global growth pattern*: This patterning combines with the overarching growth gradient of the sepal to generate cells with a wide distribution of sizes.

*Modelling*: A model in which ATML1 fluctuates stochastically and when ATML1 reaches a high concentration in G2 causes cells to stop dividing and enter endoreduplication is sufficient to produce the scattered pattern of giant cells among small cells in a growing tissue (Figure 1e, middle).

In the outer *Arabidopsis* sepal epidermis, pavement cells form in a diversity of sizes ranging from small cells to giant cells (Figures 1e and 2c,d). Plant cells are glued together by pectin in the middle lamella joining their cell walls and do not move or slip relative to one another. How does one cell decide to become a giant cell and physically enlarge, while its neighbours remain small? Screening for mutants lacking giant cells revealed that several genes involved in giant cell patterning are also involved in the epidermal specification pathway, including *Arabidopsis thaliana MERISTEM LAYER1* (*ATML1*), *HOMEODOMAIN GLABROUS11* (*HDG11*), *Arabidopsis CRINKLY4* (*ACR4*) and *DEFECTIVE KERNEL1* (*DEK1*) (Meyer et al., 2017; Roeder, Cunha, Ohno, & Meyerowitz, 2012). These epidermal specification factors are involved in specifying giant cell identity and activating differentiation and endoreduplication via the SIAMESE related CDK inhibitor LOSS OF GIANT CELLS FROM ORGANS (LGO, also known as SMR1) (Churchman et al., 2006; Kumar et al., 2015; Roeder et al., 2010; Schwarz & Roeder, 2016; Van Leene et al., 2010). Endomembrane trafficking also has a role in limiting giant cell formation (Qu et al., 2014). However, the epidermal specification pathway is active throughout the epidermis. While these findings provided mechanistic insights, they fail to address the primary question: How do cells commit to endoreduplicate or not?

Live imaging revealed that although the Homeodomain Leucine zipper class IV (HDZIP-IV) transcription factor ATML1 is expressed in all of the epidermal cells, its level fluctuates in each cell (Meyer et al., 2017). The fluctuations intersect with the cell cycle to generate the cell size pattern. In the standard mitotic cell cycle, the cell grows in G1, replicates its DNA in S phase, continues growth in G2, and divides in M phase. Endoreduplicating cells skip M phase and continue to replicate their DNA and grow (De Veylder et al., 2011). Precise quantification of ATML1 live imaging data indicated that during the G2 phase of the cell cycle, if ATML1 reaches a high concentration above a putative threshold, then the cell is likely to endoreduplicate (Figure 1e, left; Meyer et al., 2017). LGO is required for ATML1 to activate endoreduplication. Conversely if ATML1 concentration remains low throughout G2 phase, the cell is likely to divide. The ATML1 concentration during the G1 phase of the cell cycle appears to have no effect on the decision to divide or endoreduplicate. Thus, the simple local rule is that the concentration of ATML1 during G2 of the cell cycle determines whether the cell will differentiate and endoreduplicate or continue dividing (Meyer et al., 2017). This patterning combines with the overarching growth gradient of the sepal to generate cells with a wide distribution of sizes.

This local rule was simulated in a model tissue of connected growing cells. In the model ATML1 fluctuations were stochastic (Meyer et al., 2017). The cell cycle was implemented as a simple timer. Simulating the model produced a scattered pattern of giant cells among a range of smaller cells, similar to the sepal epidermis (Figure 1e, middle; Meyer et al., 2017). In the model, increasing ATML1 levels was sufficient to produce all giant cells in the sepal, which was also seen biologically when ATML1 expression was increased throughout the epidermis (Meyer et al., 2017). Likewise lowering ATML1 levels in the model replicated the *atml1* mutant which lacks giant cells.

In the model, as observed in live imaging of sepal development, the cell size pattern emerged because of the interrelationship between patterning and tissue growth (Roeder et al., 2010). The overarching growth pattern of the sepal means that cells only have a limited time to grow rapidly before they slow their growth (Hervieux et al., 2016). The cells can either use this time to divide and make more small cells or to endoreduplicate and become enlarged and polyploid (Roeder et al., 2010; Traas et al., 1998). In fact, dividing and endoreduplicating cells occur next to one another in this fast growth zone (Roeder et al., 2010). Measuring relative growth rates in dividing cell clones and non-dividing cells shows that there is no difference in growth rates (Tauriello et al., 2015). Thus, giant cells do not become enlarged by faster growth, but instead because they grow at the approximately same rate for longer without subdividing the tissue into more cells (Roeder et al., 2010; Traas et al., 1998). Similar to other emergent phenomena, timing is the key parameter to generating cells of different sizes. The earlier a cell endoreduplicates, the larger and more polyploid it becomes. Conversely, the later a cell endoreduplicates, or if it never endoreduplicates, the more cells are produced, but the smaller each of those cells is. There is a nearly balanced tradeoff between mitotic cycles and endocycles (Robinson et al., 2018).

In the current model, the decision to endoreduplicate or divide is cell autonomous. However, *ACR4* encodes a receptor kinase and *DEK1* a transmembrane calpain protease, suggesting there may be important intercellular signals involved in patterning.

Where the signal is coming from or going to is not obvious, because giant cells do not follow a classic lateral inhibition pattern of being equally spaced. Giant cells do often form in contact with one another. Thus, questions for the future include whether giant cells are randomly distributed and what is the role of intercellular communication in the cell size patterning process.

### 7.5. Emergence example 5: Gene regulatory networks in sepal initiation

*Emergent phenomenon*: The expression patterns of key regulators in the correct spatiotemporal pattern on the floral meristem (Figure 1f, right).

*Local rule*: Each regulator activates or represses other regulators according to the gene regulatory network (Figure 1f, left).

*Modelling*: Simulating the gene regulatory network was not sufficient to reproduce the spatial patterns of gene expression during early flower development. However, adding one additional hypothetical regulatory connection to each gene model, was sufficient to generate gene expression patterns closely resembling the real ones (Figure 1f, middle).

Atlases have proven useful for combining data from different studies into a more complete holistic understanding. Refahi et al., created a multiscale atlas of early flower development (https://morphonet.org) spanning the initiation on the inflorescence meristem (stage 0) through the outgrowth of sepal primordia (stage 4) (Refahi et al., 2021). Through detailed live imaging of the floral meristem and initiation of sepals, they tracked all of the cell lineages and cell growth rates. They carefully hand-annotated the gene expression patterns for 28 of the key known regulators (Figure 1f, right). One of the questions they addressed using this atlas was how the complex spatial expression patterns of each gene are established. They first tried to predict gene expression patterns at one stage based on the cell lineages from the preceding stage (i.e., assuming daughter cells express the same genes). The predictions based on lineage were fairly good at stages 1 and 4, but not at 2 or 3. Intriguingly, this result implies that regulatory interactions are most important during stages 2 and 3. Second, they simulated a Boolean gene regulatory network (GRN) that they derived from the literature (Figure 1f, left). The literature-based network did not perform better than the lineage-based network—neither sufficiently replicated the real gene expression patterns. Therefore, they assumed some of the regulatory interactions were missing from the GRN. They next tested adding a single regulatory input to each network. They tested each possible gene interaction to find the single added interaction which gave them the most accurate prediction of the gene expression pattern (Figure 1f, middle). For most of the genes, they found that adding one regulatory gene interaction significantly improved the expression pattern prediction. These added interactions are key hypotheses to be tested in the future. Finally, Refahi et al. used the fine-scale spatial relationships they obtained between gene expression patterns and growth in individual cells to better understand how genes drive morphogenesis. For example, cytokinin inhibitor AHP6 was associated with fast growth and boundary gene *CUC1* was associated with slow growth, as expected. LFY is associated with cell growth heterogeneity. This atlas is a great starting point and will become increasingly useful as the community adds additional gene expression patterns, refines the expression patterns with quantitative values, and builds more sophisticated GRNs.

## 8. Conclusion: Throwing omics at hairballs is not enough

For the past 15 years or so, biologists have become increasingly aware that reductionist experimental approaches, while certainly useful, have not been sufficient to understand the complexity of biological systems, particularly their emergent properties. This realisation led to the advent of systems biology (Trewavas, 2006). Systems biology came of age at the same time as functional-omics (genomics, proteomics, metabolomics, etc.) technologies were exploding. These technologies finally allowed us to assay the whole genome, whole transcriptome, whole proteome, and so forth, which was a big step forward towards understanding the whole system. These datasets were put together into networks in an attempt to understand them. These networks rapidly became so complex and so interconnected that they have been jokingly referred to as hairballs. Somewhere along the way, we got lost in the idea that if we just collect more and more data, we will be able to understand it all (Nurse, 2021). While exceptionally valuable, the challenge with omics data is that it tends to lose its spatial precision. This limitation in spatial and temporal resolution is also true of older technologies such as qPCR and western blots, where the tissue is ground up. Newer technologies, like single cell RNA-seq combined with 10X Visium and slide-seq are advancing us past this blockage. Yet, we find we are in need of additional non-reductionist approaches that consider the spatiotemporal relationships of cells and tissues and their mechanical interactions, which give rise to morphogenesis. In this review we have examined the computational morphodynamics approach to elucidating morphogenesis (Chickarmane et al., 2010; Roeder et al., 2011). Computational morphodynamics starts with experiments that capture the dynamics of development that help us determine the local rules that give rise to emergent properties in the morphogenetic systems. These rules are simulated in computational models to determine whether the rules are sufficient to give rise to the emergent behaviour. Clearly, as shown in Section 7.5, gene regulatory networks are important and can be integrated into this approach as the basis for modelling. As shown in the five examples, this computational morphodynamics approach has been instrumental in elucidating emergent properties generated in sepal morphogenesis.

The five examples of emergence during sepal morphogenesis discussed here reveal themes common in emergent processes. First, timing is often a key control point in emergent processes, particularly those controlling size and shape. For example, the robust size of the four sepals in the flower depends on both the nearly synchronous timing of primordium initiation and growth as well as the synchronous maturation and termination of growth (Section 7.1). Likewise, cell size is controlled by the time at which cells initiate endoreduplication; those cells endoreduplicating earliest become giant cells (Section 7.4). Second, the intersection of local cellular-scale rules with organ scale growth is critical. For example, the tapered tip shape is generated from the juxtaposition of slow growth in the tip of the sepals with fast growth at the base, which creates mechanical stress, causing the microtubules in individual cells to reorient to resist this stress (Section 7.2). Likewise, the cell size pattern of giant cells and small cells relies on the overall growth gradient, with cells exiting both endocycles and cell cycles as growth slows from the tip downward (Section 7.4). Third, shape is generated by the feedback loop between mechanical stress, microtubules reorienting to resist the stress, deposition of cellulose resisting stress, reorientation of growth, and new mechanical stress patterns associated with the new cell shapes. This feedback loop operating in different cells or different subcellular positions within

the cells contributes to the robustness of sepal shape with regard to: the formation of trichomes (Section 7.1), the tapered tip (Section 7.2), and the flattened lamina (Section 7.3). Fourth, robustness at the organ scale emerges from variation at the cellular scale (Hong et al., 2018; Zhu & Roeder, 2020). For example, variability in cellular growth creates robustness in organ size and shape through spatiotemporal averaging (Section 7.1). Likewise, variability in ATML1 expression creates the pattern of giant cells across the sepal (Section 7.4).

So far, computational morphodynamics has been used to tackle specific questions one by one. The ultimate goal is to merge these into a complete, wholistic understanding of sepal morphogenesis and combine the models into an *in silico* sepal, which could be part of a computable plant (Gor et al., 2006). However; in practice we have not even merged the existing models. One of the barriers to integration is the simple practical impediment that these models are all implemented with different code in different modelling environments, meaning that the models must be recoded to integrate them. There have been efforts to make unifying modelling languages (Keating et al., 2020), but these have yet to be widely adopted in the plant biology modelling community and may not contain all of the functionality needed for specialised applications. A more fundamental barrier is the multiscale challenge of integrating models with different levels of abstraction. The computational resources available even with the best computer clusters still set limits on the models we can simulate. Simultaneously, our biological understanding must grow to produce new hypotheses that can be encoded into models.

As yet, we have just touched the tip of the iceberg in understanding morphogenesis and many exciting questions remain to be answered. What determines the mature sepal size? Is there an organ size sensor? How does a sepal have a different size and shape than a leaf? What mechanisms control the formation of the petiole in leaves that do not occur in sepals? How do organ identity genes control morphogenesis? Most of the analysis of morphogenesis has relied on the outer epidermal cell layers due to technical limitations in imaging and image analysis leading us to ask how the underlying cell layers contribute to morphogenesis? How is sepal curvature controlled? To what extent are the lessons from *Arabidopsis* sepals generalizable to other *Arabidopsis* organs and further to other plants? The path to answer these questions lies in the integration of novel technologies that capture dynamic empirical data together with modelling to understand emergence.

## Acknowledgements

Thanks to Jeff Doyle, Kate Harline, Lilan Hong, Shuyao Kong, Erich M. Schwarz, Batthula Vijaya Lakshmi Vadde, Avilash Singh Yadav and Mingyuan Zhu for comments on the manuscript.

**Financial support.** Research in the Roeder Lab is supported by the National Institutes of Health Institute of General Medicine (grant number R01 GM134037) and the National Science Foundation Plant Fungal and Microbial Developmental Mechanisms (grant number IOS-1553030).

**Conflict of interest.** The author declares none.

**Authorship contributions.** A.H.K.R. conceived of, wrote and edited this review.

**Data availability statement.** All data are available within this review and the references therein.

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
