## [Reviewer Report]

Dear Olivier, 

Here is the long awaited review on the sepal as part of your collection of commissioned articles on key model systems for quantitative biology in plants. The theme that I highlight is emergent processes in morphogenesis of sepals, and I see that it fits with a collection the journal has started on emergence as well. When you commissioned the article there was no description of length, so I have exceeded the 5000 word limit. I hope you and the readers of Quantitative Plant Biology will enjoy.

Best wishes, 

Adrienne

---

## [Reviewer Report]

*Comments to Author*: During Arabidopsis flower development, the sepal primordium cells grow, divide, and coordinate with their neighboring cells, generating a sepal with the final size and shape. Dr. Roeder’s group has developed Arabidopsis sepals as a model system for investigating mechanisms underlying morphogenesis of lateral organs in plants. In this manuscript, entitled "Arabidopsis sepals: a model system for the emergent process of morphogenesis", the author first introduces sepal definition, staging, and growth pattern, and describes the advantages of the sepal system suitable for investigating growth and morphogenesis, because of the sepal is easy for manipulation and confocal microscopy-based live-cell imaging, and its simplicity and reproducibility of morphogenesis. Next, the author discussed the quantitative fundamentals of the sepal system for better understanding morphogenesis using computational biology and modeling. Then, the author switches to present five case studies about the emergent processes driving sepal morphogenesis. Case 1, Reactive oxygen species (ROS) regulation of robustness of sepal size and shape. Case 2, A microtubule-dependent mechanical feedback loop restricts sepal tip and shape.

Case 3, A model explaining formation of a laminar plant organ such as a flat leaf blade or sepal, in which the mechanical response of microtubules occurs in the inner cell walls, but not in the outer periclinal cell walls, generates a flat laminar form. Case 4, Cell size patterning in the sepal epidermis. Case 5, Gene regulatory networks in sepal initiation. Finally, the author concludes with some challenges and proposes outstanding questions for the future studies. In my opinion, this manuscript is a well-written, comprehensive review in the field, and thus very suited for publication in Quantitative Plant Biology.

---

## [Reviewer Report]

*Comments to Author*: This is a very nicely written and illustrated review that both provides a comprehensive view of sepal structure and interest as a model system and demonstrates how computational morphodynamics can allow understanding emergent properties.

The 5 examples are very clearly discussed and, starting from them, a larger perspective is drawn. Many challenging questions are indicated.

I have two suggestions that may help going even further in the discussion.

1. So far computational morphodynamics was used to tackle specific questions one by one. How far are we in starting to merge these different models to provide a global model that would allow integrating GRN and growth to reconstruct sepal morphogenesis. Are there major limitations for this, and if, are they more biological (do we miss some biological understanding for some key processes), computational?

2. The approaches described here are mostly cell-centered with emerging properties occurring at the organ level. Several decades ago, the relevance of the cell theory for plants was a matter of debate (see eg Kaplan, D.R., and Hagemann, W. (1991). The relationship of cell and organism in vascular plants: Are cells the building blocks of plant form? Bioscience 41, 693-703). This debate proposed to address plant morphogenesis starting from the organ level. I am wondering how the progress of computational morphodynamics and the new concepts generated intersect with such an alternative organismal view.

Minor comments

- L29. Not sure what the difference between shape and form is.

- L20. The link between PIN orientation and mechanical interactions could be detailed more.

- Paragraph starting at line 111. I am not sure that describing the ABCE model is absolutely required for this review. It may be removed to have a more focused review. Alternatively, discussing how these homeotic genes may impact morphogenesis of the sepal could be interesting for this review.

- L144. Robustness of sepal morphology is stressed here, leading to idea that analysis can be performed on a single plant. But what is known about the relative variability within and between individuals. This question is particularly relevant as further below (line 214) it is mentioned that later sepals lack trichomes

- Line400. Indicate why the katanin mutant inhibits mechanical feedback. Similarly describe better the modification occuring in the spiral2 mutant.

---

## [Reviewer Report]

*Comments to Author*: Both reviewers found the review is well-written and a timely summary of the emerging field. Reviewer 2 commented on the possible integration of separate models, and on the idea of organ-centered understanding of morphogenesis. I would like to recommend the author to take these comments into consideration while revising the manuscript.

---

## [Reviewer Report]

Dear Yuling and Olivier, I thank the reviewers for their thoughtful comments, which I have addressed as detailed in the response to reviewers. Best wishes, Adrienne